# Brain responsivity to emotional faces differs in men and women with and without a history of alcohol use disorder

Marlene Oscar-Berman[1,2,3], Susan Mosher Ruiz[1,2], Ksenija Marinkovic[4], Mary M. Valmas[1,2,3], Gordon J. Harris[3], Kayle S. Sawyer [ID][1,2,3,5]*

1 Psychology Research Service, VA Boston Healthcare System, Boston, MA, United States of America, 2 Department of Anatomy & Neurobiology, Boston University School of Medicine, Boston, MA, United States of America, 3 Department of Radiology, Massachusetts General Hospital, Boston, MA, United States of America, 4 Department of Psychology, San Diego State University, San Diego, CA, United States of America, 5 Sawyer Scientific, LLC, Boston, MA, United States of America

* kslays@bu.edu

**Data Availability Statement:** The data used to generate the line graph figures are publicly available, included in the supplementary data. Other data contain potentially identifying or

## Abstract

Inclusion of women in research on Alcohol Use Disorder (AUD) has shown that gender differences contribute to unique profiles of cognitive, emotional, and neuropsychological dysfunction. We employed functional magnetic resonance imaging (fMRI) of abstinent individuals with a history of AUD (21 women [AUDw], 21 men [AUDm]) and demographically similar non-AUD control (NC) participants without AUD (21 women [NCw], 21 men [NCm]) to explore how gender and AUD interact to influence brain responses during emotional processing and memory. Participants completed a delayed match-to-sample emotional face memory fMRI task, and brain activation contrasts between a fixation stimulus and pictures of emotional face elicited a similar overall pattern of activation for all four groups. Significant Group by Gender interactions revealed two activation clusters. A cluster in an anterior portion of the middle and superior temporal gyrus, elicited lower activation to the fixation stimulus than to faces for the AUDw as compared to the NCw; that abnormality was more pronounced than the one observed for men. Another cluster in the medial portion of the superior frontal cortex elicited higher activation to the faces by AUDm than NCm, a difference that was more evident than the one observed for women. Together, these findings have added new evidence of AUD-related gender differences in neural responses to facial expressions of emotion.

## Introduction

Chronic long-term Alcohol Use Disorder (AUD), also referred to as "alcoholism," is a harmful condition [1, 2] that has been associated with deficits in emotion and memory, including memory for the emotional expressions of faces [3–6]. In addition to its effects on memory for facial emotions, AUD also has been associated with impairments in the processing of facial emotional expressions [7–11], and these effects can endure after months of sobriety [12, 13].

sensitive information from a vulnerable population, including direct and indirect identifiers necessary for analyses such as face images contained in MRI scans, medical history, and gender. The Boston University Medical Campus Institutional Review Board may be contacted at 617-358-5372 or medirb@bu.edu.

**Funding:** This work was supported by funds from the National Institute on Alcohol Abuse and Alcoholism (NIAAA; https://www.niaaa.nih.gov/) of the National Institutes of Health US Department of Health and Human Services under Award Numbers R01AA007112 and K05AA00219 awarded to M.O. B., as well as R01AA016624 and K01AA13402 awarded to K.M.; US Department of Veterans Affairs Clinical Science Research and Development (https://www.research.va.gov/services/csrd/) grant I01CX000326 awarded to M.O.B.; National Center for Advancing Translational Sciences of the National Institutes of Health US Department of Health and Human Services under Award Numbers 1S10RR023401, 1S10RR019307, 1S10RR023043, and 1UL1TR001430. K.S.S. is an employee of Sawyer Scientific, LLC. The funders provided support in the form of salaries for authors M.O.B., S.M.R., K.M., M.M.V., G.J.H., and K.S.S., but did not have any role in the study design, data collection and analysis, decision to publish, or preparation of the manuscript. The specific roles of these authors are articulated in the 'author contributions' section. The content is solely the responsibility of the authors and does not necessarily represent the official views of the National Institutes of Health, the U.S. Department of Veterans Affairs, or the United States Government.

**Competing interests:** K.S.S. is an employee of Sawyer Scientific, LLC. This does not alter our adherence to PLOS ONE policies on sharing data and materials. There are no patents, products in development or marketed products to declare.

Brain regions associated with the encoding of face identity and emotional expressions have been established [14], and research is beginning to indicate how the activity of these regions varies in conjunction with AUD. In the present study, we explored face encoding in particular, which relies upon prefrontal cortex, amygdala, hippocampus, fusiform, and lateral parietal regions [15–17]. Processing of emotional facial expressions and identity appear to be partially functionally segregated [18]. That is, attention to the identity of faces and to the emotions expressed on them, influences the way in which networks are utilized. Attention to face identity typically activates fusiform and inferior temporal areas, and attention to the emotional expression has been shown to activate superior temporal, amygdala, putamen, insula, cingulate, and inferior frontal regions [19–24]. Functional magnetic resonance imaging (fMRI) studies of facial emotion processing among healthy adults have reported activation in a widespread network of brain areas. These areas include the fusiform gyrus, lateral occipital gyrus, superior temporal sulcus, inferior frontal gyrus, insula, orbitofrontal cortex, basal ganglia, and the amygdala [25–28]. These were the amygdala, fusiform gyrus, hippocampus, parahippocampal gyrus, intraparietal sulcus, orbitofrontal cortex, superior temporal gyrus, and superior temporal sulcus.

Facial identity also has been examined in the context of gender [29]. For example, men have been reported to show stronger activation than women in several frontal and temporal brain regions [30, 31], suggesting potential gender dimorphism in the processing of facial emotional expressions. The higher activation of these regions in men could be related to other factors observed to vary with emotion, such as hormones, genetics, and culture [29]. In addition to differentiating how facial emotion is processed, gender also influences the ways in which alcoholism impacts the brain, and further, how emotion, gender, and alcoholism interact.

Historically, AUD has afflicted more men than women, but prevalence for women has increased such that they are approaching similar rates [5, 32–36]. Much of what has been learned about the long-term effects of alcoholism has been based upon research that focused on men, or has not differentiated the results obtained from different genders. There exist clear differences in how alcohol affects men and women cognitively and physiologically [37] and in how they progress from social to problem drinkers [34, 38]. AUD-related gender effects on brain structure also have been described, involving white matter volume [39–42], morphometry of the brain reward system [43], and cerebellar subregional volumes [44]. Additionally, we have reported AUD-related gender-dimorphic effects in multiple functional domains including emotional processing [45], personality [46], and drinking motives [5, 47]. However, functional magnetic resonance imaging (fMRI) studies of AUD-related gender differences in the brain during performance of tasks involving emotion are rare [45].

The present study sought to characterize abnormalities in neural activation among abstinent participants with AUD, through analysis of BOLD responses to photographs of faces that varied in emotional expressions. We were particularly interested in observing fMRI activation in brain regions that subserve face processing, memory encoding, and emotions, and additionally, in characterizing how these effects differ between AUD men (AUDm) and AUD women (AUDw), as compared to non-AUD control men (NCm) and women (NCw). When forming our specific a priori hypothesis with regard to AUD-related gender differences in response to emotional stimuli, we noted that the literature reported mixed findings. Across brain regions, previous literature has described abnormalities in several directions, with women evidencing greater brain activativation in response to emotional stimuli, men having more brain activity, or having undetectable gender differences [5, 11, 45, 48–50]. Based upon prior research in our laboratory, wherein brain regions of AUDw (compared to NCw) were found to be overactive in response to highly charged emotional stimuli [45], in the present study we hypothesized

that the pattern of AUD-related activation abnormalities would differ for men and women, and the brain responses to emotional faces by the AUDw would be hyperactive. Additionally, we expected to replicate prior results [45, 51] showing widespread lower responses in AUDm than NCm.

## Methods

### Participants

Participants included 42 abstinent individuals with a history of long-term AUD (21 AUDw) and 42 controls (21 NCw) (Table 1). Participants were recruited within the Boston area through advertisements placed in public spaces (*e.g.*, hospitals, churches, stores), newspapers, and internet websites. This research was approved by the Boston Medical Center/Boston University School of Medicine Institutional Review Board (#H24686), VA Boston Healthcare System Institutional Review Board (#1017 and #1018), and the Partners Healthcare System Human Research Institutional Review Board (#2000P001891). Participants provided written informed consent for participation in the study. Participants gave written informed consent prior to participation, and were compensated for their time.

Selection procedures included an initial structured telephone interview to determine age, level of education, health history, and history of alcohol and drug use. It should be noted that we used the term "gender," because we did not assess biological characteristics such as sex chromosomes or reproductive anatomy. Included participants were right-handed, had normal or corrected-to-normal vision, and spoke English as a first language. Participants were interviewed further to determine use of alcohol and other drugs, including prescription drugs that would affect the central nervous system. Current drug use excepting nicotine and caffeine was cause for exclusion. Criteria for exclusion also included history of liver disease, epilepsy, head trauma resulting in loss of consciousness for 15 minutes or more, HIV, symptoms of Korsakoff's syndrome or dementia, and schizophrenia. Additionally, individuals who failed MRI screening (e.g., metal implants and obesity) were excluded.

Neuropsychological testing was conducted at the Department of Veterans Affairs (VA) Boston Healthcare System facility. Participants completed a medical history interview, vision test, handedness questionnaire [52], Hamilton Rating Scale for Depression [53] and the Diagnostic and Statistical Manual (DSM-IV) Diagnostic Interview Schedule [54]. Participants also were administered the Wechsler Adult Intelligence Scale (WAIS-III or WAIS-IV) and the Wechsler Memory Scale (WMS-III or WMS-IV) [55, 56].

The participants with AUD met DSM-IV criteria for alcohol abuse or dependence, and consumed 21 or more alcoholic drinks per week for a total of five or more years. The extent of alcohol use was assessed by calculating Quantity Frequency Index (QFI) scores [57]. QFI approximates the amount, type, and frequency of alcohol consumption either over the last six months (control participants), or over the six months preceding cessation of drinking (participants with AUD) to yield an estimate of ounces of ethanol per day; this is similar to the number of drinks consumed per day. The participants with AUD were abstinent for at least four weeks before the scan date. Control participants who had consumed 15 or more drinks per week for any length of time, including binge drinking, were excluded.

### Functional imaging task

Participants were given a delayed match-to-sample memory task in an MRI scanner, whereby they were asked to encode two faces that had one of three emotional valences (positive, negative, or neutral), followed by one of three types of distractor cues (alcoholic beverage, nonalcoholic beverage, or a scrambled image) and a probe stimulus (an emotional face) for

**Table 1. Participant characteristics.**

| | AUD GROUP | | | CONTROL GROUP | | |
|---|---|---|---|---|---|---|
| | **All** | **Women** | **Men** | **All** | **Women** | **Men** |
| | *n = 42* | *n = 21* | *n = 21* | *n = 42* | *n = 21* | *n = 21* |
| **Age[a] (years)** | | | | | | |
| *mean* | 53.9 | 53.4 | 54.4 | 53.9 | 57.7 | 50.2 |
| *standard deviation* | 11.0 | 11.4 | 10.8 | 12.4 | 13.6 | 10.1 |
| *range* | 26.5–76.7 | 26.5–73.0 | 26.6–76.7 | 25.8–76.9 | 25.8–76.9 | 29.0–69.6 |
| **Education[b] (years)** | | | | | | |
| *mean* | 14.7 | 15.3 | 14.1 | 15.5 | 15.6 | 15.4 |
| *standard deviation* | 2.0 | 2.0 | 1.9 | 2.0 | 2.3 | 1.6 |
| *range* | 12–19 | 12–19 | 12–18 | 12–19 | 12–20 | 12–18 |
| **WAIS-III Full Scale IQ** | | | | | | |
| *mean* | 110.3 | 110.1 | 110.5 | 111.6 | 111.2 | 112.0 |
| *standard deviation* | 15.0 | 14.2 | 16.0 | 16.3 | 19.3 | 13.1 |
| *range* | 72–140 | 72–137 | 81–140 | 79–152 | 79–142 | 90–152 |
| **WMS-III IMI** | | | | | | |
| *mean* | 109.7 | 114.4 | 104.7 | 111.9 | 114.8 | 109.0 |
| *standard deviation* | 16.6 | 18.3 | 13.4 | 16.9 | 16.4 | 17.4 |
| *range* | 63–144 | 63–144 | 82–130 | 80–146 | 84–138 | 80–146 |
| **WMS-III DMI** | | | | | | |
| *mean* | 112.6 | 116.7 | 108.3 | 111.8 | 113.5 | 110.1 |
| *standard deviation* | 17.3 | 20.4 | 12.5 | 16.0 | 14.9 | 17.2 |
| *range* | 52–140 | 52–140 | 86–132 | 83–150 | 83–140 | 84–150 |
| **Duration of Heavy Drinking[f] (years)** | | | | | | |
| *mean* | 17.4 | 14.3 | 20.5 | NA | NA | NA |
| *standard deviation* | 7.7 | 5.2 | 8.5 | | | |
| *range* | 5.0–35.0 | 6.0–25.0 | 5.0–35.0 | | | |
| **Quantity Frequency Index[cde] (ounces ethanol/day)** | | | | | | |
| *mean* | 11.2 | 8.7 | 13.7 | 0.3 | 0.2 | 0.4 |
| *standard deviation* | 8.8 | 5.8 | 10.5 | 0.6 | 0.5 | 0.7 |
| *range* | 2.7–38.4 | 2.7–28.1 | 4.5–38.4 | 0.0–2.6 | 0.0–2.4 | 0.0–2.6 |
| **Length of Sobriety[cde] (years)** | | | | | | |
| *Mean* | 8.3 | 10.6 | 5.9 | 2.1 | 3.6 | 0.5 |
| *standard deviation* | 10.3 | 11.1 | 8.8 | 6.4 | 8.5 | 1.3 |
| *range* | 0.1–32.3 | 0.1–32.1 | 0.1–32.3 | 0.002–29.2 | 0.002–29.2 | 0.002–5.1 |
| **Hamilton Rating Scale for Depression[g]** | | | | | | |
| *mean* | 3.5 | 4.9 | 2.2 | 2.4 | 3.1 | 1.8 |
| *standard deviation* | 4.2 | 4.1 | 4.0 | 2.8 | 3.3 | 2.1 |
| *range* | 0–18 | 0–17 | 0–18 | 0–12 | 0–12 | 0–8 |

Participant Characteristics ($p < 0.05$)

[a]NCw > NCm

[b]NCm > AUDm

[c]AUD > NC

[d]AUDm > NCm

[e]AUDw > NCw

[f]AUDm > AUDw

[g]AUDw > AUDm

Abbreviations: WAIS—Wechsler Adult Intelligence Scale; WMS—Wechsler Memory Scale; IMI—Immediate Memory Index; DMI—Delayed Memory Index.

comparison (Fig 1 and S1 Fig). This task was chosen specifically because AUD has been associated with deficits in emotional functions [5], and with face memory in particular [51], but women were not included in prior research. The faces were intended to display happy (positive), neutral, and sad (negative) expressions. The faces were shown in grayscale and were taken from a set of faces used in a previous study [51]. Face stimuli were displayed simultaneously for three seconds, followed by an asterisk (*) for one second. Participants were asked to maintain these faces in memory while distractor stimuli were shown (for three seconds), followed by an asterisk (*) for one second, immediately followed by a probe face (shown for two seconds) to assess memory for face identity, and ending with a variable-length fixation stimulus (+++; for 2 to 22 seconds, average 10 seconds). It should be noted that the results pertaining to the fMRI encoding portion of the task were analyzed separately from the distractor and probe portions of the task; behavioral and fMRI findings based upon analyses of the distractor and probe portions comprise a separate research report. The participants responded to the probe face with their right index or middle fingers, and psychophysiological recordings were taken from the left hand [58]. The event-related design used nine six-minute runs with 18 trials per run, for a total of 162 trials. There were 54 trials for each emotional face valence. Counterbalancing and inter-trial intervals were calculated with optseq2 [59].

## Image acquisition, processing, and statistical analyses

The MRI brain images were acquired by a 3T scanner and processed using Freesurfer (https://surfer.nmr.mgh.harvard.edu/), as summarized in this paragraph and presented in detail in the following paragraphs. Structural scans were acquired at 1 x 1 x 1.33 mm resolution, and fMRI scans were acquired with 3.125 x 3.125 x 5 mm resolution and a TR of 2530 ms. Cortical surfaces were reconstructed from structural scans. First-level and group-level smoothing were set to 5 mm and 8 mm, and the cluster threshold was set to $p < 0.05$ with a primary threshold of $p < 0.001$. Contrasts for the facial emotion conditions included: positive, negative, and neutral vs. fixation; and the three contrasts between the conditions. Intergroup comparisons were made among the subgroups (men, women, AUD, NC, AUDm, NCm, AUDw, and NCw, as well as Group by Gender interactions. We selected eight a priori regions of interest (ROI) because their separate locations had previously been established to be involved in emotion and face processing, and we investigated group differences for each region. BOLD responses were entered as dependent variables in a repeated-measures ANOVA, with between-group factors of Group and Gender, and the within-subjects factor of facial Emotion.

Imaging was conducted at the Massachusetts General Hospital, Charlestown, MA. Data were acquired on a 3 Tesla Siemens (Erlangen, Germany) MAGNETOM Trio Tim MRI scanner with a 12-channel head coil. Sagittal T1-weighted MP-RAGE scans (TR = 2530 ms, TE = 3.39 ms, flip angle = 7˚, FOV = 256 mm, slice thickness = 1.33 mm, slices = 128, matrix = 256 x 192) were collected for all subjects. Echo planar fMRI scans were acquired axially with 5 mm slice thickness and 3.125 x 3.125 mm in-plane resolution (64 x 64 matrix), allowing for whole brain coverage (32 interleaved slices, TR = 2 s, TE = 30 ms, flip angle = 90˚). Within each six-minute run, 180 T2*-weighted volumes were obtained. Functional volumes were auto-aligned to the anterior/posterior commissure line to ensure a similar slice prescription was employed across participants. Prospective Acquisition Correction (3D-PACE) was applied during acquisition of the functional volumes to minimize the influence of participants' body motion [60]. A laptop running Presentation version 11.2 software (NeuroBehavioral Systems, Albany, CA) was used for visual presentation of the experimental stimuli and collection of participants' responses. Stimuli were back-projected onto a screen at

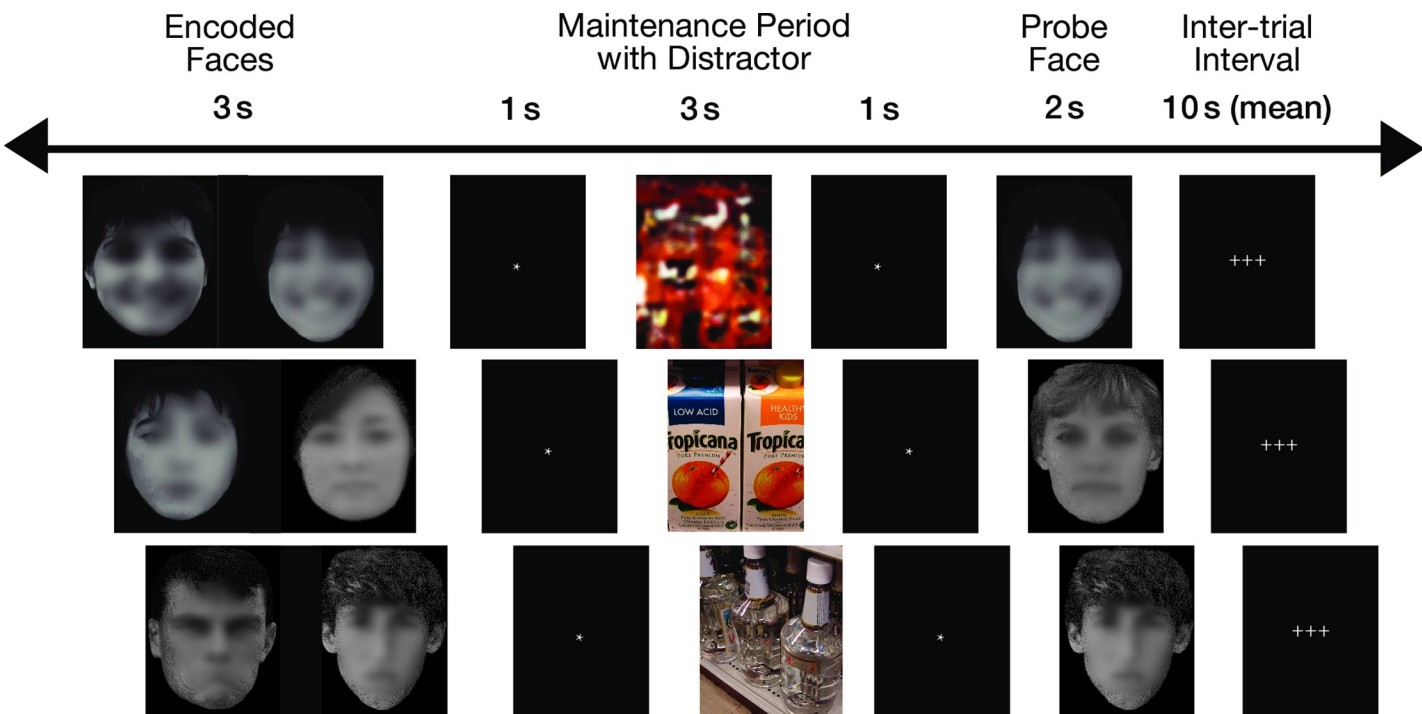

**Fig 1. Task presented during functional neuroimaging.** Two faces were presented simultaneously for 3 seconds, followed by an asterisk for one second. Next, a distractor was presented for 3 seconds, followed by an asterisk for 1 second. The probe face immediately followed, during which the subjects had been trained to respond with a button press with either their index or middle finger to indicate whether the probe face matched the encoded face. Three crosses served as the fixation stimulus and inter-trial interval, which lasted from 2 to 22 seconds (mean 10 seconds). The faces in this figure have been blurred to mask the identities of the individuals, but the research participants saw the original unblurred photographs.

the back of the scanner bore and viewed by participants through a mirror mounted to the head coil. Participants wore earplugs to attenuate scanner noise.

Cortical surfaces were reconstructed using Freesurfer version 4.5.0 (http://surfer.nmr.mgh. harvard.edu) to obtain segmentation labels [61, 62] along with white matter and exterior cortical surfaces [63]. These were visually inspected on each coronal slice for every subject, and manual interventions (*e.g.*, white matter volume corrections) were made as needed. In order to delineate small regions distinguished by gyri and sulci, we used the Destrieux atlas [64] for segmentation and parcellation of anatomical ROI in the functional analyses.

Effects of Group, Gender, and Emotion on BOLD signal were evaluated using a whole-brain cluster analysis, as well as ROI analyses. Processing of the functional data was performed using FreeSurfer Functional Analysis Stream (FS-FAST) (http://surfer.nmr.mgh.harvard.edu/ ), SPSS Version 17.0 (IBM, Chicago, IL, USA), and custom Matlab scripts (The MathWorks, Natick, MA).

Preprocessing of functional images for first-level (individual subject) analyses included motion correction, intensity normalization, and spatial smoothing with a 5-mm Gaussian convolution kernel at full-width half-maximum, as specified with FS-FAST. BOLD response was estimated using a Finite Impulse Response (FIR) model, which allows for estimation of the time course of activity (percent signal change for a given condition) within a voxel, vertex, or ROI for the entire trial period. For each condition, estimates of signal intensity were calculated for 2 pre-trial and 10 post-trial onset time points (TRs), for a total analysis window of 24 seconds. Motion correction parameters calculated during alignment of the functional images were entered into the analysis as external regressors. Alignment of the T2*-weighted functional

images with T1-weighted structural volumes was accomplished through an automated boundary-based registration procedure [65]. These automated alignments were manually inspected to ensure accuracy.

For contrasts between experimental conditions, statistical maps were generated via FS-FAST for each research participant. Contrasts for the facial emotion conditions included: (1) positive faces vs. fixation, (2) negative faces vs. fixation, (3) neutral faces vs. fixation, (4) positive faces vs. negative faces, (5) positive faces vs. neutral faces, and (6) negative faces vs. neutral faces. Analyses of each of these contrasts included removal of prestimulus differences between the contrasted conditions by averaging the first three time points (two pre-trial and one post-trial onset) for each condition and subtracting this mean from each time point for that condition. Time points summed for inclusion in each contrast were chosen to reflect peak stimulus-related activity: FIR estimates of hemodynamic responses to the emotion effects were examined during the time period of 2–10 seconds post emotional face onset.

Second-level (group) analyses on cortical regions were accomplished using a surface-based morphing procedure for intersubject alignment and statistics [66], as performed with FS-FAST. Group-averaged signal intensities during each experimental condition relative to fixation were calculated using the general linear model in spherical space for cortical regions, and were mapped onto the canonical cortical surface *fsaverage*, generating group-level weighted random-effects *t*-statistic maps. The same procedure was performed for the volume with a subcortical mask (MNI305 space; 2mm isotropic voxels). An 8 mm full-width half-maximum smoothing kernel was employed for all group and intergroup maps.

Intergroup comparison *t*-statistic maps were generated to examine between-group effects by contrasting: (1) AUD vs. NC, (2) AUDm vs. NCm, (3) AUDw vs. NCw, (4) AUDm vs. AUDw, (5) NCm vs. NCw, and (6) men vs. women. Additionally, Group by Gender interaction maps for each contrast were calculated.

We selected multiple correction procedures and thresholds that balance Type I and Type II error levels. For the statistical maps, cluster-level corrections for multiple comparisons were applied using the permutation procedure implemented by the mri_glmfit-sim procedure included in FS-FAST 6.0 [67], with 1,000 permutations. We chose to use permutation testing to identify clusters, because it provides the best control for Type I error, as compared to correction procedures that incorporate Gaussian Random Field theory [67, 68]. A primary (vertex- and voxel-wise) threshold of $p < 0.001$ was applied, and clusters with *p*-values $< 0.05$ (further corrected for analysis of three spaces, left, right, and volume) were selected. A primary threshold of $p < 0.001$ is stringent, and therefore eliminates smaller, more regionally specific clusters, inflating Type II error. However, to test our specific hypotheses, we conducted separate group comparisons between the four subgroups, in addition to examining the Group by Gender interaction; when viewed as a single family, these additional comparisons inflate Type I error. As described in the Limitations, the direct assessment of group differences has an advantage over using a factorial ANOVA in that it could identify clusters in areas without the interaction.

The cluster regions were identified by the location of each cluster's peak vertex or voxel according to the Destrieux atlas [64], but the clusters reported can be understood to span multiple functional regions [68]. That is, they are not limited to a single region, as reported by the maximal vertex or voxel.

Eight anatomically-defined ROI were selected for the emotional faces BOLD analyses to include regions identified *a priori* that are known to be involved in the recognition of emotions, and in visual processing and memory encoding of human faces. These were the amygdala, fusiform gyrus, hippocampus, parahippocampal gyrus, intraparietal sulcus, orbitofrontal cortex, superior temporal gyrus, and superior temporal sulcus. Left and right hemisphere regions were analyzed separately.

Statistical preprocessing and time course visualization of ROI data were performed using custom scripts written for Matlab version 7.4.0. Signal intensity for each region was averaged across all vertices (or voxels) included in the region for each condition on the individual participant level. To compute percent signal change for each participant within an ROI, signal estimate per condition and time point was divided by the average baseline activity for that participant in the same manner as for the statistical maps. Group and Group-by-Gender averages of the normalized time courses were computed for each condition, and were visualized by plotting the percent signal change for each condition at each time point (TR) of the trial.

For the ROI analyses, percent signal changes of the BOLD signal within each ROI were entered as dependent variables into repeated-measures ANOVA models with between-group factors of Group (alcoholic or control) and Gender (men or women) and within-subjects factor of facial Emotion (positive, negative, or neutral).

## Results

### Participant characteristics

Table 1 summarizes means, standard deviations, and ranges of participant demographics, IQ and memory test scores, and drinking variables. The AUD and NC groups did not differ significantly by age (mean age 54 years), and although NCw were older than NCm, control groups did not differ significantly from the respective AUD groups of the same gender. AUDm had one year less education than NCm. Groups did not differ significantly on WAIS-III Full Scale IQ scores. While AUDw had higher Hamilton Rating Scale for Depression scores than AUDm, the average scores for all four subgroups (AUDm, AUDw, NCm, and NCw) were low (all means below 5, whereas mild depression threshold is 8), suggesting that depression contributed little to our observed gender differences. The AUD participants drank 11 drinks per day on average, had a mean duration of 17 years of heavy drinking, and were sober for an average of eight years. The differences between AUDm and AUDw for QFI and length of sobriety were not significant, but the DHD for the AUDm was significantly longer by 6.2 years compared to the AUDw. Five NCm and two NCw reported being lifetime abstainers, and one NCm reported three years of binge drinking in college, and sporadic drinking thereafter.

### Neuroimaging cluster analyses

Group-level cluster analyses of each facial emotion condition vs. fixation yielded clusters too large to be described in an anatomically-relevant way with a single peak location. Therefore, these data are summarized qualitatively in the text, and they are illustrated in Fig 2. Group-level clusters are reported in S1 Table for contrasts with fixation (Positive vs. Fixation, Negative vs. Fixation, Neutral vs. Fixation) and in Table 2 for emotion contrasts (positive vs. negative, positive vs. neutral, and negative vs. neutral). For voxel-wise analyses of contrasts with fixation, no clusters were significant. Intergroup clusters are reported for each emotion condition vs. fixation (Table 3), and no intergroup clusters for emotion contrasts were significant.

The AUD and NC groups of both genders utilized a distributed network of cortical brain regions to process faces of all three emotional valences as compared to the fixation stimulus. As an example, Fig 2 shows the *t*-statistic cluster map of the contrast of positive vs. fixation displayed on the lateral surface; the medial and lateral views of the negative vs. fixation and neutral vs. fixation are shown in S2 through S5 Figs. Subcortical volume-based *t*-statistic cluster maps are shown in S6 through S9 Figs. Several *face-activated* regions were more responsive when participants viewed the face stimuli than during the fixation condition: dorsolateral prefrontal cortex, motor cortex, anterior insula, inferior temporal cortex (including fusiform),

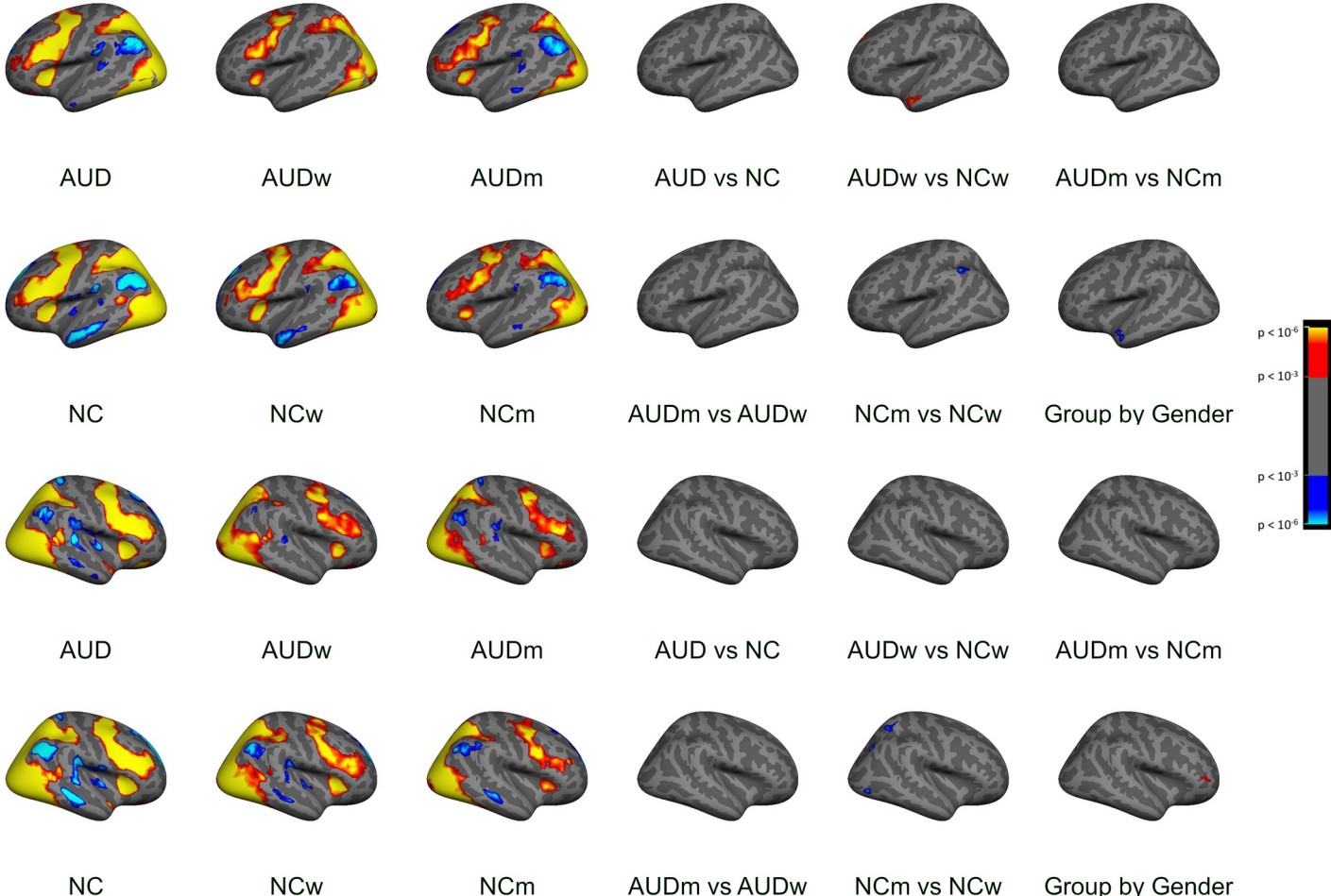

**Fig 2. Cortical surface cluster maps: Positive vs. fixation, lateral.** The left three columns show group maps, and the right three columns show group comparisons. The top two rows represent the left hemisphere, and the bottom two rows represent the right hemisphere. The clusters in this figure had a vertex wise threshold of $p <$ .001 with a minimum cluster size of 100 mm$^2$. This can result in more clusters being visible than are listed in Table 3 and S1 Table, wherein numbers were derived using permutation testing (cluster threshold $p <$ .05, further corrected for analyses of left, right, and volume spaces).

parietal cortex, the occipital lobes, and limbic structures. A different set of *fixation-activated* regions was more active during fixation than during the face conditions, including regions that are typically more active during rest than during attentionally-demanding cognitive tasks [69].

As was observed for the contrast of faces vs. fixation, results for the contrasts between emotions revealed several regions, but the clusters were of smaller spatial extent and less consistent across participant groups. Significant clusters are summarized in Table 2 and shown in S10 through S15 Figs. Positive faces elicited the least activation, as compared to both neutral and negative faces. These effects were significant primarily in regions of the frontal lobes, although additional regions identified were in the occipital and temporal lobes.

Clusters of significant between-group differences on each emotion condition vs. fixation are described in Table 3. The results were complicated and differed by brain region, contrast direction, and group comparison. For positive faces, significant clusters were identified within all lobes of the brain. The group contrast directions for all of these clusters indicated greater activation values in the subgroup of NCw compared to the AUDw and NCm, and the effects were observed across both fixation-activated and face-activated regions. It should be noted

**Table 2. Emotion whole brain group cluster analysis: Positive vs. negative, positive vs. neutral, negative vs. neutral.**

| Negative Faces vs. Positive Faces | | | | | | | |
|---|---|---|---|---|---|---|---|
| Structure at Peak Vertex | Size (mm$^2$) | MNIX | MNIY | MNIZ | CWP | Group | Contrast Direction |
| Right Superior Temporal Sulcus | 780.0 | 48.4 | -45.3 | 6.2 | 0.003 | NC | NEG > POS |
| Right Anterior Occipital Sulcus | 409.0 | 40.1 | -68.1 | 0.7 | 0.006 | NC | NEG > POS |
| Right Precentral Gyrus | 392.7 | 45.5 | 1.7 | 45.7 | 0.006 | NC | NEG > POS |
| Right Occipital Pole | 172.2 | 21.3 | -89.5 | -6.4 | 0.015 | NC | NEG > POS |
| Right Inferior Frontal Gyrus Pars Triangularis | 129.7 | 52.3 | 29.8 | 5.8 | 0.027 | NC | NEG > POS |
| Right Superior Temporal Sulcus | 469.9 | 48.4 | -41.4 | -2.5 | 0.009 | WOMEN | NEG > POS |
| Right Occipital Pole | 156.6 | 19.5 | -90.2 | -6.5 | 0.036 | WOMEN | NEG > POS |
| Left Middle Anterior Cingulate | 135.7 | -11.9 | 19.3 | 30.1 | 0.018 | AUDm | NEG > POS |
| Right Superior Temporal Sulcus | 363.1 | 46.8 | -45.1 | 7.8 | 0.003 | NCm | NEG > POS |
| Right Inferior Frontal Sulcus | 312.3 | 36.5 | 19.0 | 25.9 | 0.003 | MEN | NEG > POS |
| Neutral Faces vs. Positive Faces | | | | | | | |
| Structure at Peak Vertex | Size (mm$^2$) | MNIX | MNIY | MNIZ | CWP | Group | Contrast Direction |
| Left Superior Frontal Gyrus | 156.2 | -8.9 | 0.8 | 55.3 | 0.012 | AUD | NEU > POS |
| Left Middle Frontal Gyrus | 222.4 | -39.1 | 38.4 | 27.3 | 0.003 | NC | NEU > POS |
| Right Superior Frontal Gyrus | 307.6 | 6.8 | 1.2 | 65.6 | 0.006 | NC | NEU > POS |
| Right Middle Frontal Sulcus | 142.1 | 24.7 | 44.1 | 27.8 | 0.033 | NC | NEU > POS |
| Left Fronto-marginal Gyrus (of Wernicke) and Sulcus | 88.82 | -24.2 | 51.9 | -5.3 | 0.048 | NC | NEU > POS |
| Right Superior Frontal Gyrus | 85.09 | 16.4 | -2.4 | 67.0 | 0.027 | NCw | NEU > POS |
| Left Mid-Anterior Cingulate | 217.8 | -10.7 | 12.9 | 49.9 | 0.006 | WOMEN | NEU > POS |
| Right Superior Frontal Gyrus | 145.7 | 15.4 | -2.5 | 67.6 | 0.015 | WOMEN | NEU > POS |
| Left Superior Frontal Gyrus | 140.2 | -10.1 | 3.7 | 67.1 | 0.021 | WOMEN | NEU > POS |
| Right Superior Temporal Sulcus | 388.5 | 48.3 | -47.8 | 22.0 | 0.003 | NCm | NEU > POS |
| Right Middle Frontal Gyrus | 222.1 | 32.9 | 46.7 | 16.7 | 0.009 | NCm | NEU > POS |
| Right Occipital Pole | 149.6 | 11.8 | -88.0 | 0.7 | 0.021 | NCm | NEU > POS |
| Structure at Peak Voxel | Size (mm$^3$) | | | | | | |
| Right Putamen | 3472 | 18.0 | 13.0 | -5.0 | 0.003 | NC | NEU > POS |
| Negative Faces vs. Neutral Faces | | | | | | | |
| Structure at Peak Vertex | Size (mm$^2$) | MNIX | MNIY | MNIZ | CWP | Group | Contrast Direction |
| Right Anterior Transverse Collateral Sulcus | 230.2 | 38.6 | -19.3 | -27.0 | 0.015 | AUD | NEG > NEU |
| Right Middle Temporal Gyrus | 161.9 | 56.1 | -57.7 | 1.6 | 0.018 | AUD | NEG > NEU |

Coordinates are presented for peak vertices and voxels within significant clusters of activation. Minimum significance for all vertices (or voxels) within a cluster were $p = 0.001$. Clusters were selected using permutation testing at $p < 0.05$. For context, S1 Table presents significant clusters for contrasts with fixation (Positive vs. Fixation, Negative vs. Fixation, Neutral vs. Fixation). S6 through S9 Figs show the corresponding subcortical cluster maps, and S10 through S15 Figs show the corresponding cortical cluster maps. The clusters reported can be understood to span multiple functional regions [68]. That is, they are not limited to a single region, as reported by the maximal vertex or voxel. Abbreviations: MNIX, MNIY, MNIZ—Montreal Neurological Institute 305 Atlas coordinates of maximum vertex; CWP—Cluster-wise $p$ value.

that the colors and corresponding directions of effects in the figures are shown by calculating results for faces minus fixation, whereas in Table 3, the directions of absolute values are presented. For example, in the left superior temporal gyrus, Table 3 shows the NCw had greater fixation activation than AUDw, and in Fig 2, the NCw are shown to have lower face activation than AUDw (although it may be awkward to think of "negative face activation").

Significant Group by Gender interaction effects for temporal and frontal regions were driven by the lower activation of NCm than NCw, while AUDm had similar or greater activation than AUDw. The negative faces revealed a pattern of group differences that encompassed

**Table 3. Emotion whole brain intergroup cluster analysis: Positive vs. fixation, negative vs. fixation, neutral vs. fixation, positive vs. negative, positive vs. neutral, negative vs. neutral.**

**Positive Faces vs. Fixation**

| Structure at Peak Vertex | Size (mm²) | MNIX | MNIY | MNIZ | CWP | Group Comparison | Condition Contrast |
|---|---|---|---|---|---|---|---|
| Left Superior Temporal Gyrus | 494.7 | -48.4 | -51.2 | 44.2 | 0.003 | NCw > AUDw | FIX > POS |
| Left Supramarginal Gyrus | 483.2 | -50.3 | -50.8 | 44.1 | 0.006 | NCw > NCm | POS > FIX |
| Right Intraparietal Sulcus | 265.0 | 26.6 | -54.3 | 52.6 | 0.036 | NCw > NCm | POS > FIX |
| Left Angular Gyrus | 644.8 | -48.4 | -51.2 | 44.2 | 0.003 | WOMEN > MEN | POS > FIX |
| Left Superior Temporal Gyrus | 323.6 | -50.4 | 9.5 | -17.6 | 0.024 | Group by Gender: | FIX > POS |
|  |  |  |  |  |  | NCw > AUDw; |  |
|  |  |  |  |  |  | AUDm ≈ NCm |  |

**Negative Faces vs. Fixation**

| Structure at Peak Vertex | Size (mm²) | MNIX | MNIY | MNIZ | CWP | Group Comparison | Condition Contrast |
|---|---|---|---|---|---|---|---|
| Left Superior Frontal Gyrus | 267.9 | -6.6 | 33.8 | 49.8 | 0.042 | AUDm > NCm | NEG > FIX |
| Left Supramarginal Gyrus | 447.7 | -50.3 | -50.8 | 44.1 | 0.015 | NCw > NCm | NEG > FIX |
| Left Angular Gyrus | 378.5 | -48.4 | -51.2 | 44.2 | 0.018 | MEN > WOMEN | FIX > NEG |
| Left Superior Frontal Gyrus | 319.5 | -7.0 | 26.9 | 40.8 | 0.030 | Group by Gender: | NEG > FIX |
|  |  |  |  |  |  | AUDm > NCm; |  |
|  |  |  |  |  |  | AUDw ≈ NCw |  |

**Neutral Faces vs. Fixation**

| Structure at Peak Vertex | Size (mm²) | MNIX | MNIY | MNIZ | CWP | Group Comparison | Condition Contrast |
|---|---|---|---|---|---|---|---|
| Right Anterior Cingulate | 285.9 | 11.5 | 52.8 | 4.1 | 0.027 | NCw > NCm | FIX > NEU |
| Left Supramarginal Gyrus | 307.3 | -48.9 | -50.4 | 44.3 | 0.027 | WOMEN > MEN | NEU > FIX |

Coordinates are presented for peak voxels within significant clusters of activation. Minimum between-group significance for all vertices (or voxels) within a cluster was $p = 0.001$. Clusters were selected using permutation testing at $p < 0.05$. No clusters were significant for contrasts between facial emotions (Positive vs. Negative, Positive vs. Neutral, Negative vs. Neutral). Fig 2 and S1 through S5 Figs show the corresponding cortical cluster maps. As noted in the text, the colors and corresponding directions of effects in the figures are shown by calculating results for faces minus fixation, whereas in Table 3, the directions of absolute values are presented. The clusters reported can be understood to span multiple functional regions [68]. That is, they are not limited to a single region, as reported by the maximal vertex or voxel. Abbreviations: MNIX, MNIY, MNIZ—Montreal Neurological Institute 305 Atlas coordinates of maximum vertex; CWP—Cluster-wise $p$-value.

many brain regions. For supramarginal regions, NCw had greater values than NCm. As was found for the positive faces, a frontal cluster was identified where a significant Group by Gender interaction was driven by stronger negative vs. fixation contrast values obtained from AUDm than NCm, and a smaller difference was observed between AUDw and NCw. The neutral faces revealed two clusters, with NCw having greater contrasts than NCm in the fixation-activated anterior cingulate cortex (S5 Fig). No clusters were identified with significant group differences for contrasts between emotional face conditions (Positive vs. Negative, Positive vs. Neutral, Negative vs. Neutral).

### Neuroimaging region of interest analyses

Results of repeated-measures ANOVAs examining between-subjects effects of Group and Gender and within-subjects effects of facial Emotion on BOLD percent signal change within each ROI are reported below. Means and standard deviations represent the percent signal change across each ROI for the time period of 2 to 10 seconds post-face stimulus onset.

**Intraparietal sulcus.** A significant main effect of Group was found for left intraparietal sulcus activation during encoding of the emotional faces ($F = 4.172$, $p = 0.044$). As can be seen in Fig 3, activation to the faces was substantially higher than to the fixation for the AUD and NC groups alike. The NC group had more activity (vs. fixation) in this region during face

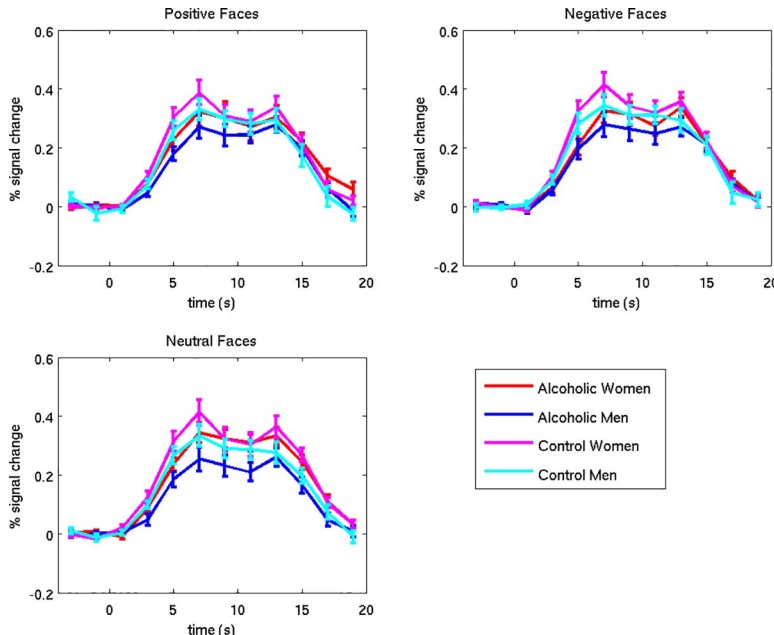

**Fig 3. Left intraparietal sulcus.** Error bars represent the standard error of the mean. The analysis window used to examine the encoded faces was 2 to 10 seconds. The first peak (at approximately 7 seconds) represents brain activity resulting from the encoded faces. The activity following that peak (longer than 10 seconds) shows responses to the distractor and probe stimuli, to be described in a future research report.

encoding (*Mean* = 0.27, *SD* = 0.12) than did the AUD group (*Mean* = 0.21, *SD* = 0.13). Activity in the right intraparietal sulcus did not vary significantly by Group, Gender, or facial Emotion.

**Hippocampus.** Overall, hippocampal activation in response to the encoded faces (vs. fixation) was negligible for all groups (Fig 4). Although there was a significant Emotion by Group by Gender interaction for the left hippocampus ($F_{1,80}$ = 4.005, $p$ = 0.049), intergroup comparisons were not significant. The directions of the effects were as follows: For NCm, activation was higher for positive faces vs. fixation (*Mean* = 0.04, *SD* = 0.09) than for neutral faces vs. fixation (*Mean* = 0.01, *SD* = 0.10) and for negative faces vs. fixation (*Mean* = 0.02, *SD* = 0.07); the same means were observed for AUDw. This pattern appeared stronger than that observed for the other subgroups (ALCm *Means*: positive = -0.01, neutral = -0.01, negative = -0.01; NCw *Means*: positive = -0.02, neutral = 0.01, negative = 0.01). Activity in the right hippocampus did not vary significantly by Group, Gender, or Emotion.

**Amygdala, fusiform, orbitofrontal cortex, parahippocampal cortex, superior temporal gyrus, and superior temporal sulcus.** Activity in these regions did not vary significantly by Group, Gender, or Emotion.

In summary, for ROI analyses, we observed higher responsivity during face encoding in the NC group compared to the AUD group, in the left intraparietal sulcus, a region that has been identified as playing an important role in focusing attention to enhance working memory [70]. In the left hippocampus, a region involved in memory, a significant interaction of Emotion, Group, and Gender, indicated that the NCm activated this region more to positive faces than to neutral faces, as compared to the other subgroups. However, the magnitudes of these effects were small.

## Discussion

In this study, we used an fMRI task to explore how gender and AUD interacted to influence brain activation levels during emotional processing and memory. Here, we discuss the findings

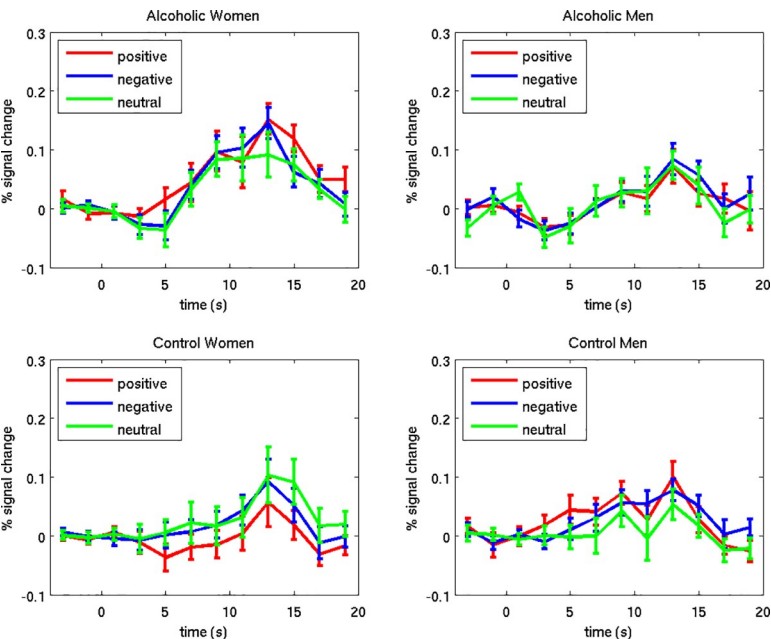

**Fig 4. Left hippocampus.** Error bars represent standard error of the mean. The percent signal change represents brain activity resulting from presentation of the task stimuli. The analysis window used to examine the encoded faces was 2 to 10 seconds. Brain activity following that period shows responses to the distractor and probe stimuli, to be described in a future research report.

that were similar for the AUD and NC groups of both genders, that is, greater cortical responsivity to negative and neutral facial expressions than to positive faces in frontal, temporal, and occipital regions. We also discuss the finding that the AUD cohort evidenced greater fusiform activation in response to negative than to positive facial expressions. Next, as a background with which to frame our results regarding gender differences, we discuss the activation clusters identified from comparisons between AUD and NC groups. We then highlight the clusters found with Group by Gender interaction effects, and the implications of the lower activation levels in NCm compared to NCw in comparison to those between AUDm and AUDw. Finally, we note that results for two of the eight ROI we examined had significant group comparisons: The left intraparietal sulcus showed lower activation by the AUD group than the NC group, and the left hippocampus showed a complex Group by Gender by Emotion interaction.

## Responsivity to facial expressions

The whole-brain group-level cluster analyses comparing activation among facial emotion conditions showed that participants had higher responses both to negative and to neutral faces than to positive faces in frontal cortex, superior temporal sulcus, and occipital pole. While it is unclear why neutral faces would elicit higher activation than positive faces, a meta-analysis [14] showed that negative faces were widely reported to evoke strong activation. Consistent with our results, the frontal cortex is involved in recognition of facial emotions [51]. Further, enhanced activity during emotional face recognition in response to disgusted, angry, or fearful expressions relative to neutral expressions, has been reported in inferior frontal cortex [71]. In addition to the involvement of frontal regions in facial processing, the superior temporal sulcus has been implicated in differentiating emotional facial expressions in nonclinical populations [18, 19, 72]. This pattern of face activation was consistent with the AUD group's responses to negative faces, which were greater than to neutral faces in the collateral sulcus

(adjacent to the anterior fusiform gyrus). While the fusiform is more commonly implicated in processing facial identity information [18], some studies also have implicated the role of the collateral sulcus [19, 72]. Interestingly, a review of studies that focused on valence effects of face processing [73] reported a developmental progression of biases that favor "more efficient processing of positive over negative faces," wherein children more easily process positive faces, but with age, this evolves into a more efficient processing of negative facial expressions.

## Intergroup clusters

Examination of whole-brain intergroup cluster comparisons revealed gender differences among controls wherein the difference in responses to emotional faces vs. fixation was larger for NCw than NCm in the supramarginal gyrus and intraparietal sulcus, adjacent areas that may have important roles in emotional processing [74]. A medial portion of the superior frontal gyrus was identified wherein AUDm had more activation to negative faces as compared to fixation than did NCm. However, this cluster should be interpreted in the context of a significant interaction between group and gender for the same region.

## Interactions between group and gender

The NC participants demonstrated more reliable gender differences in neural responses than did the AUD group, and widespread neural responses to these stimuli were more pronounced in NCw than in NCm. By comparison, gender differences in the AUD group were muted. The Group by Gender interaction effects indicated stronger responses to emotional faces by AUDm than NCm in the left superior frontal gyrus, a significantly more pronounced group difference than the one observed for women (S3 Fig). In an anterior portion of the left superior temporal gyrus (extending to the middle temporal gyrus) that was more highly activated by fixation than positive faces, we observed a similar interaction effect wherein NCw had greater fixation-related activation than AUDw, a group difference that was more pronounced than the one observed for men (Fig 2).

We had hypothesized that (1) AUD-related abnormalities would differ for men and women; (2) AUDw would show hyperactivation to emotional faces; and (3) AUDm would show hypoactivation compared to NCm. Although we did observe gender differences in AUD-related abnormalities, the other hypotheses were not confirmed, namely we found that activation by AUDw to emotional faces was hypoactive, and activation by the AUDm was hyperactive. Possible explanations for these findings derive from neuropsychiatric literature. For example, women are more likely than men to be diagnosed with depression, anxiety disorders, and eating disorders [75]. Therefore, alcohol consumption by AUDw might reflect a predisposition to abnormal regional activation in either direction (hyper- or hypo-responsivity) that leads to depression, anxiety disorders, and eating disorders. In contrast, AUDm might consume alcohol to enhance pleasurable activities [76], although gender differences in drinking motives are not always observed [47]. Whether cause or effect, AUD in men and women could become part of a vicious cycle that serves as a coping mechanism and also reinforces abnormalities in brain responsivity.

## Regions of interest

The ROI analyses examining effects of Group, Gender, and Emotional face valence yielded few significant results. Overall, in the left intraparietal sulcus, a region that has been singularly identified as playing an important role in focusing attention to enhance working memory [70], we found high levels of activation to the faces in both the NC and AUD groups. Moreover, the responses were greater for the NC group than for the AUD group, thereby supporting other

evidence of working memory deficits in AUD populations [5]. We also found that the AUD group showed hypoactivation of the left intraparietal sulcus when encoding the identity of the emotional face stimuli. Of interest, Majerus et al. [77] reported that during the performance of a face encoding task by healthy young adults, activation of the left intraparietal sulcus showed preferential functional connectivity with right temporal, inferior parietal, and medial frontal areas involved in detailed face processing. The authors noted that these results supported an attentional account of left intraparietal sulcus involvement in visual short term memory, and highlighted the importance of the left intraparietal sulcus as an attentional modulator in a variety of short term memory tasks. The findings of Majerus and colleagues are consistent with our interpretation that the AUD group's hypoactivation in the intraparietal sulcus reflects memory and attentional impairments in association with chronic abuse of alcohol [3–6]. The significant interaction of Emotion, Group, and Gender we found in the left hippocampus, showing that NCm and AUDw activated this region more for positive faces than for neutral faces, was not observed for NCw and AUDm. However, as compared to fixation, activation of the hippocampus to the encoded faces was negligible in all groups, which constrains the interpretability of these results.

## Limitations

As noted in the Methods, the encoded face fMRI data were acquired in the context of a task that measured the influence of distractor cues. The task contained a memory component measuring participants' response accuracies and reaction times to probe faces (not to encoded faces). Therefore, interference from the distractor images may have influenced brain-activation or behavior. However, the distractor stimuli were equally distributed following each of the three types of emotional face valence conditions in order to ameliorate any differential impact of the distractors. Results obtained from the influences of the distractor and probe stimuli on emotional processing will be presented in a future report.

While we performed multiple comparisons correction procedures by using permutation testing, we conducted ROI analyses without correcting for multiple comparisons. Each ROI was selected individually, because each of them was independently derived from results of previous literature. However, this approach also could be considered in the context of a family of comparisons with an elevated false positive error rate.

For our ROI results, we employed a traditional approach to statistical analyses, by examining interaction effects, followed by determining the significance of group differences. Instead of using a factorial ANOVA to assess the significant activation clusters with subsequent within-cluster post-hocs (that is, combining group comparisons with interaction effects for vertex-wise and voxel-wise analyses), we assessed the activation clusters for the group differences directly across all vertices and voxels. Even though this approach engenders additional comparisons and increases the likelihood of making Type I errors, it has the benefit of identifying clusters in locations other than those where interactions are present. We used a primary (vertex- or voxel-wise) threshold of $p < 0.001$ for each group comparison. By considering these analyses to be a family of seven comparisons, the Bonferroni-adjusted family-wise primary threshold would be $p < 0.007$.

Because this study employed cross-sectional observations, we cannot determine whether hazardous drinking caused, or resulted from, dysregulated emotional reactivity. Further, because we had limited information about the potentially confounding variable of smoking status, we did not include it in our analyses. The effects of abstinence from smoking have been associated with increased emotional reactivity in response to negative stimuli [78] and research additionally has implicated interactions between smoking and alcoholism [79, 80], so smoking effects could have influenced our results.

Finally, our AUD subjects were abstinent for a minimum of four weeks but an average of 8.3 years. This wide range of sobriety speaks to the persistent nature of emotion processing deficits in AUD populations, and whether such deficits recover differently over the course of short- and long-term abstinence in men and women [13]. Moreover, we believe that the alcohol consumption and abstinence characteristics of our AUD cohort are representative of the national population [36], thereby improving generalizability of our results. Additionally, the topic of persistence vs. recovery remains a promising direction for future studies, for example, with analyses of brain activity in relation to length of sobriety and gender for the AUD group [81, 82]. The average length of sobriety was longer for AUDm than for AUDw, which might have influenced the gender differences we observed. Since there were no abstinence values for the NC group, the variable could not be used as a covariate in an analysis of group differences.

## Supporting information

**S1 Fig. Cortical surface cluster maps: Positive vs. fixation, medial.** The left three columns show group maps, and the right three columns show group comparisons. The top two rows represent the left hemisphere, and the bottom two rows represent the right hemisphere. The clusters in this figure had a vertex wise threshold of $p < .001$ with a minimum cluster size of 100 mm$^2$. This can result in more clusters being visible than are listed in Table 3 and S1 Table, wherein numbers were derived using permutation testing (cluster threshold $p < .05$, further corrected for analyses of left, right, and volume spaces).
(PNG)

**S2 Fig. Cortical surface cluster maps: Negative vs. fixation, lateral.** The left three columns show group maps, and the right three columns show group comparisons. The top two rows represent the left hemisphere, and the bottom two rows represent the right hemisphere. The clusters in this figure had a vertex wise threshold of $p < .001$ with a minimum cluster size of 100 mm$^2$. This can result in more clusters being visible than are listed in Table 3 and S1 Table, wherein numbers were derived using permutation testing (cluster threshold $p < .05$, further corrected for analyses of left, right, and volume spaces).
(PNG)

**S3 Fig. Cortical surface cluster maps: Negative vs. fixation, medial.** The left three columns show group maps, and the right three columns show group comparisons. The top two rows represent the left hemisphere, and the bottom two rows represent the right hemisphere. The clusters in this figure had a vertex wise threshold of $p < .001$ with a minimum cluster size of 100 mm$^2$. This can result in more clusters being visible than are listed in Table 3 and S1 Table, wherein numbers were derived using permutation testing (cluster threshold $p < .05$, further corrected for analyses of left, right, and volume spaces).
(PNG)

**S4 Fig. Cortical surface cluster maps: Neutral vs. fixation, lateral.** The left three columns show group maps, and the right three columns show group comparisons. The top two rows represent the left hemisphere, and the bottom two rows represent the right hemisphere. The clusters in this figure had a vertex wise threshold of $p < .001$ with a minimum cluster size of 100 mm$^2$. This can result in more clusters being visible than are listed in Table 3 and S1 Table, wherein numbers were derived using permutation testing (cluster threshold $p < .05$, further corrected for analyses of left, right, and volume spaces).
(PNG)

**S5 Fig. Cortical surface cluster maps: Neutral vs. fixation, medial.** The left three columns show group maps, and the right three columns show group comparisons. The top two rows represent the left hemisphere, and the bottom two rows represent the right hemisphere. The clusters in this figure had a vertex wise threshold of $p < .001$ with a minimum cluster size of 100 mm$^2$. This can result in more clusters being visible than are listed in Table 3 and S1 Table, wherein numbers were derived using permutation testing (cluster threshold $p < .05$, further corrected for analyses of left, right, and volume spaces).
(PNG)

**S6 Fig. Subcortical volume cluster maps: AUDw, positive faces vs. fixation.** Cluster-corrected at $p < .001$ with minimum cluster size 300 mm$^3$. Shown in neurological convention (left brain on the left side of the image).
(PNG)

**S7 Fig. Subcortical volume cluster maps: AUDm, positive faces vs. fixation.** Cluster-corrected at $p < .001$ with minimum cluster size 300 mm$^3$. Shown in neurological convention (left brain on the left side of the image).
(PNG)

**S8 Fig. Subcortical volume cluster maps: NCw, positive faces vs. fixation.** Cluster-corrected at $p < .001$ with minimum cluster size 300 mm$^3$. Shown in neurological convention (left brain on the left side of the image).
(PNG)

**S9 Fig. Subcortical volume cluster maps: NCm, positive faces vs. fixation.** Cluster-corrected at $p < .001$ with minimum cluster size 300 mm$^3$. Shown in neurological convention (left brain on the left side of the image).
(PNG)

**S10 Fig. Cortical surface cluster maps: Positive vs. negative, lateral.** The left three columns show group maps, and the right three columns show group comparisons. The top two rows represent the left hemisphere, and the bottom two rows represent the right hemisphere. The clusters in this figure had a vertex wise threshold of $p < .001$ with a minimum cluster size of 100 mm$^2$. This can result in more clusters being visible than are listed in Table 2, wherein numbers were derived using permutation testing (cluster threshold $p < .05$, further corrected for analyses of left, right, and volume spaces).
(PNG)

**S11 Fig. Cortical surface cluster maps: Positive vs. negative, medial.** The left three columns show group maps, and the right three columns show group comparisons. The top two rows represent the left hemisphere, and the bottom two rows represent the right hemisphere. The clusters in this figure had a vertex wise threshold of $p < .001$ with a minimum cluster size of 100 mm$^2$. This can result in more clusters being visible than are listed in Table 2, wherein numbers were derived using permutation testing (cluster threshold $p < .05$, further corrected for analyses of left, right, and volume spaces).
(PNG)

**S12 Fig. Cortical surface cluster maps: Positive vs. neutral, lateral.** The left three columns show group maps, and the right three columns show group comparisons. The top two rows represent the left hemisphere, and the bottom two rows represent the right hemisphere. The clusters in this figure had a vertex wise threshold of $p < .001$ with a minimum cluster size of 100 mm$^2$. This can result in more clusters being visible than are listed in Table 2, wherein

numbers were derived using permutation testing (cluster threshold $p < .05$, further corrected for analyses of left, right, and volume spaces).
(PNG)

**S13 Fig. Cortical surface cluster maps: Positive vs. neutral, medial.** The left three columns show group maps, and the right three columns show group comparisons. The top two rows represent the left hemisphere, and the bottom two rows represent the right hemisphere. The clusters in this figure had a vertex wise threshold of $p < .001$ with a minimum cluster size of 100 mm$^2$. This can result in more clusters being visible than are listed in Table 2, wherein numbers were derived using permutation testing (cluster threshold $p < .05$, further corrected for analyses of left, right, and volume spaces).
(PNG)

**S14 Fig. Cortical surface cluster maps: Negative vs. neutral, lateral.** The left three columns show group maps, and the right three columns show group comparisons. The top two rows represent the left hemisphere, and the bottom two rows represent the right hemisphere. The clusters in this figure had a vertex wise threshold of $p < .001$ with a minimum cluster size of 100 mm$^2$. This can result in more clusters being visible than are listed in Table 2, wherein numbers were derived using permutation testing (cluster threshold $p < .05$, further corrected for analyses of left, right, and volume spaces).
(PNG)

**S15 Fig. Cortical surface cluster maps: Negative vs. neutral, medial.** The left three columns show group maps, and the right three columns show group comparisons. The top two rows represent the left hemisphere, and the bottom two rows represent the right hemisphere. The clusters in this figure had a vertex wise threshold of $p < .001$ with a minimum cluster size of 100 mm$^2$. This can result in more clusters being visible than are listed in Table 2, wherein numbers were derived using permutation testing (cluster threshold $p < .05$, further corrected for analyses of left, right, and volume spaces).
(PNG)

**S1 Table. Emotion whole brain group vertex cluster analysis: Positive vs. fixation, negative vs. fixation, neutral vs. fixation.** Coordinates are presented for peak vertices within significant clusters of activation for surface analyses of the left and right hemispheres. Minimum significance for all vertices within a cluster were $p = 0.001$. Clusters were selected using permutation testing at $p < 0.05$. For context, Table 2 presents significant intergroup clusters for these contrasts; Fig 2 and S1 through S5 Figs show the corresponding cortical cluster maps. The clusters reported can be understood to span multiple functional regions [68]. That is, they are not limited to a single region, as reported by the maximal vertex. Abbreviations: ClusterNo —The cluster number within the analysis; Max—Maximum -log($p$ value) within the cluster; VtxMax—Vertex number for peak vertex; MNIX, MNIY, MNIZ—Montreal Neurological Institute 305 Atlas coordinates of maximum vertex; CWP—Cluster-wise $p$ value; CWPLow, CWPHi—90% confidence interval for the CWP; NVtxs—number of vertices in cluster; Annot —Destrieux annotation for peak vertex.
(CSV)

**S1 File.**
(CSV)

**S2 File.**
(CSV)

## Acknowledgments

The authors thank Elinor Artsy, Sheeva Azma, Zoe Gravitz, Doug Greve, Anne-Mette Guld-berg, Maaria Kemppainen, Steve Lehar, Diane Merritt, Alan Poey, Elizabeth Rickenbacher, Trinity Urban, and Robert Zondervan for assistance with manuscript preparation and recruit-ment, assessment, analysis, or neuroimaging of the research participants. We also greatly appreciate the helpful suggestions provided by the reviewers. Finally, we would like to acknowledge the role of the research participants for making this study possible.

## Author Contributions

**Conceptualization:** Marlene Oscar-Berman, Susan Mosher Ruiz, Ksenija Marinkovic, Gordon J. Harris, Kayle S. Sawyer.

**Data curation:** Susan Mosher Ruiz, Kayle S. Sawyer.

**Formal analysis:** Susan Mosher Ruiz, Kayle S. Sawyer.

**Funding acquisition:** Marlene Oscar-Berman, Ksenija Marinkovic.

**Investigation:** Susan Mosher Ruiz, Mary M. Valmas, Kayle S. Sawyer.

**Methodology:** Susan Mosher Ruiz, Kayle S. Sawyer.

**Project administration:** Marlene Oscar-Berman, Susan Mosher Ruiz, Gordon J. Harris, Kayle S. Sawyer.

**Resources:** Marlene Oscar-Berman.

**Software:** Susan Mosher Ruiz, Kayle S. Sawyer.

**Supervision:** Marlene Oscar-Berman, Susan Mosher Ruiz, Ksenija Marinkovic, Gordon J. Harris, Kayle S. Sawyer.

**Visualization:** Susan Mosher Ruiz, Kayle S. Sawyer.

**Writing – original draft:** Marlene Oscar-Berman, Susan Mosher Ruiz, Kayle S. Sawyer.

**Writing – review & editing:** Marlene Oscar-Berman, Ksenija Marinkovic, Kayle S. Sawyer.

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
