## [Decision Letter · Decision Letter 0]

19 Feb 2020

PONE-D-20-00671

Brain responsivity to emotional faces differs in men and women with and without a history of alcohol use disorder

PLOS ONE

Dear Dr. Kayle S Sawyer,

Thank you for submitting your manuscript to PLOS ONE. After careful consideration, we feel that it has merit but does not fully meet PLOS ONE’s publication criteria as it currently stands. Therefore, we invite you to submit a revised version of the manuscript that addresses the points raised during the review process.

We would appreciate receiving your revised manuscript by Apr 04 2020 11:59PM. To enhance the reproducibility of your results, we recommend that if applicable you deposit your laboratory protocols in protocols.io, where a protocol can be assigned its own identifier (DOI) such that it can be cited independently in the future. For instructions see: http://journals.plos.org/plosone/s/submission-guidelines#loc-laboratory-protocols

We look forward to receiving your revised manuscript.

Kind regards,

Wi Hoon Jung, PhD

Academic Editor

PLOS ONE

Journal Requirements:

1. We noticed you have some minor occurrence of overlapping text with the following previous work, which needs to be addressed:

https://elifesciences.org/articles/41723

The text that needs to be addressed involves the limitations section.

In your revision ensure you cite all your sources (including your own works), and quote or rephrase any duplicated text outside the methods section. Further consideration is dependent on these concerns being addressed.

2. Thank you for including the following funding statement within the acknowledgements section; "This work was supported by funds from the US Department of Veterans Affairs Clinical Science Research and Development grant I01CX000326; the National Institute on Alcohol Abuse and Alcoholism (NIAAA) of the National Institutes of Health, US Department of Health and Human Services, under Award Numbers R01AA07112, R01AA016624, K05AA00219, and K01AA13402; and shared instrumentation grants 1S10RR023401, 1S10RR019307, and 1S10RR023043 from the National Center for Research Resources (now National Center for Advancing Translational Sciences) at the Athinoula A. Martinos Center, Massachusetts General Hospital."

3. 

We note that Supplemental Figure [1] includes an image of a [patient / participant / in the study]. 

Additional Editor Comments (if provided):

The reviewers note some important points that need clarification, such as re-checking some information in the figures, statistical thresholds and so on. The authors need to address them.

Reviewers' comments:

Reviewer's Responses to Questions

**Comments to the Author**

1. Is the manuscript technically sound, and do the data support the conclusions?

Reviewer #1: Yes

Reviewer #2: Partly

2. Has the statistical analysis been performed appropriately and rigorously? 

Reviewer #1: Yes

Reviewer #2: Yes

3. Have the authors made all data underlying the findings in their manuscript fully available?

Reviewer #1: Yes

Reviewer #2: Yes

4. Is the manuscript presented in an intelligible fashion and written in standard English?

Reviewer #1: Yes

Reviewer #2: Yes

5. Review Comments to the Author

Reviewer #1: This study replicates a previous fMRI study but includes women in order to begin to better understand sexual dimorphism in the brain with respect to the effects of alcoholism on emotional processing and memory. A total of 42 individuals with alcohol use disorder (AUD) and 42 healthy controls were included: half of each group (n=21) was women. All participants completed a delayed match to sample emotional face memory fMRI task. Regions activated by the task were similar to those previously identified in the literature, especially for normal controls. Within both groups, sex differences emerged. In the healthy controls, sex differences were more pronounced and neural responses were more widespread in women than men. Sex differences in the AUD group were attenuated and sometimes in the opposite direction than those observed in healthy controls.

The groups were well matched and the activation patterns were generally similar across the groups, lending assurance to replicability. The writing was clear and succinct.

Some things to address:

AUD subjects were abstinent for a minimum of 4 weeks but an average of 8 years. This wide range of sobriety supports persisting deficits in emotional and memory processing, but also in a heterogenous group, could length of sobriety have contributed to sex differences. Can length of sobriety be used as a co-variate in the analysis?

Why was the Destrieux atlas used? Can some sort of justification be presented?

AUDw had higher Hamilton Depression Scale (HDS) scores than AUDm. Can the sex differences be accounted for by depression? Can the HDS score be used as a covariate?

Positive faces elicited the least activation, as compared to both neutral and negative faces. Can the authors discuss this result in a bit more detail?

Reviewer #2: This manuscript examines emotional responsiveness to faces in adults with alcohol use disorder. This is an interesting question where further work is needed, as many studies to date have been underpowered to address this question. Also, more clearly investigating gender differences in AUD could help with more personalized assessment/treatment of AUD. However, there are several outstanding methodological and analytic questions that should be addressed before I recommend this manuscript for publication. I also believe that providing some clear hypotheses (with reasoning for these hypotheses) in the introduction would help with framing the overall manuscript and with data interpretation.

Introduction

Page 4:

In paragraph 2 on Page 4, the authors transition from talking about the neurobiology underlying facial identity versus expression recognition in the previous paragraph to talking about gender differences in AUD. As a reader, I found this transition to be somewhat confusing and would suggest reframing paragraph 2 to begin by discussing the neurobiology underlying face and emotion processing and then transitioning to the potential implications for gender differences.

Page 4-5:

The authors do not seem to offer any a priori hypotheses regarding gender differences in facial processing in AUD and only state that they wished to “characterize abnormalities in neural activation among abstinent participants with AUD.” I think that stating a clear hypothesis about what the authors expected to find would be helpful in terms of framing this study within the context of the current literature and guiding the subsequent analyses and discussion.

Methods

Page 8:

“The following a priori ROI were selected due to their previously established involvement in the emotion and face processing: amygdala, fusiform gyrus, hippocampus, parahippocampal gyrus, intraparietal sulcus, orbitofrontal cortex, superior temporal gyrus, and superior temporal sulcus.”

Did the authors correct for multiple comparisons across multiple a priori ROI’s? If not, could the authors clarify why they did not correct for multiple comparisons?

Page 8:

“For surface analysis, the cluster threshold was set to 100 mm2 contiguous voxels of p <

0.001 for each contrast; for volume analysis, the minimum volume of contiguous voxels was 300

mm3.”

How were these thresholds determined? Using AFNI clustsim or a similar tool? Also, what do 100mm2 and 300 mm3 translate to in terms of voxels?

Page 9:

“For each participant, contrasts for the facial emotion conditions included: (1) positive faces vs. fixation, (2) negative faces vs. fixation, (3) neutral faces vs. fixation, (4) positive faces vs. negative faces, (5) positive faces vs. neutral faces, and (6) negative faces vs. neutral faces. Intergroup comparisons were: (1) AUD vs. NC, (2) AUDm vs. NCm, (3) AUDw vs. NCw, (4) AUDm vs. AUDw, (5) NCm vs. NCw, (6) men vs. women, and (7) Group by Gender interactions.”

For the cluster-level analyses, can the authors explain why they chose to implement individual t-tests for each contrast/intergroup comparison? In my view it would be simpler to model this as a 2 (Group: AUD vs NC) x 2 (Gender: Male vs Female) x 3 (Facial Expression: Positive, Negative, Neutral) repeated measures ANOVA and then submit any significant F-tests to post-hoc testing.

I would also recommend moving the supplemental methods section and supplemental Figure 1 to the main text to give more details regarding the MRI data collection, analysis, and the behavioral paradigm.

Results

Page 9:

In the participants section, could the authors state whether there were any differences in alcohol quantity/frequency between males and females, particularly within the AUD group? I see that these differences are listed in Table 1, but it would be helpful to have these results in the main text.

Page 10:

“A different set of fixation-activated regions was more active during fixation than

during the face conditions, forming the network known as the default mode network, because

those regions typically are more active during rest than during attentionally-demanding cognitive tasks (Buckner et al., 2008). The regions making up this network include the superior frontal cortex, medial prefrontal cortex, medial temporal lobe structures, the middle temporal gyrus, the posterior cingulate cortex plus precuneus, and the angular gyrus.”

I would recommend removing this part regarding the default mode network as the main contrasts of interest primarily involve regions that are responsive to facial emotions and DMN findings are not discussed in the discussion section.

Page 11-12:

“The three clusters that were identified with significant group differences for contrasts between emotional face conditions had peak voxels contained within the left and right amygdala (872 mm3 and 384 mm3, respectively), and left hypothalamus (464 mm3).”

Could the authors clarify which clusters these are or display them in a figure? I do not see the any clusters listed as amygdala or hypothalamus in Table 3.

Page 12-13:

In the Neuroimaging Region of Interest Analyses section, did the intraparietal sulcus and/or the hippocampus show greater activation for faces relative to fixation within the ROI? If not, interpreting group differences may be difficult.

Finally- were there any group differences in behavior (either response times or accuracy) on the task? This would provide a bit more context in which to interpret the neuroimaging findings.

Discussion

Page 13-14:

The first paragraph of the discussion seems to discuss the main task findings; while this is important information, I think it would be helpful to have a short paragraph at the very beginning of the discussion summarizing the results in order to provide the reader a road map for the discussion.

Figures/Tables

It would be helpful to have a figure showing some of the cluster-level group differences that are listed out in Table 3. In particular, it would be helpful to have a figure that not only shows each group but also directly displays the cluster-level group differences, particularly within the amygdala/hypothalamus (as mentioned in the Results section on Pages 11-12).

6. PLOS authors have the option to publish the peer review history of their article (what does this mean?). If published, this will include your full peer review and any attached files.

Reviewer #1: No

Reviewer #2: No

---

## [Author Response · Author response to Decision Letter 0]

28 Aug 2020

Please see attached response to reviewers.

---

## [Decision Letter · Decision Letter 1]

6 Oct 2020

PONE-D-20-00671R1

Brain responsivity to emotional faces differs in men and women with and without a history of alcohol use disorder

PLOS ONE

Dear Dr. Sawyer,

Thank you for submitting your manuscript to PLOS ONE. After careful consideration, we feel that it has merit but does not fully meet PLOS ONE’s publication criteria as it currently stands. Therefore, we invite you to submit a revised version of the manuscript that addresses the points raised during the review process.

We look forward to receiving your revised manuscript.

Kind regards,

Wi Hoon Jung, PhD

Academic Editor

PLOS ONE

Reviewers' comments:

Reviewer's Responses to Questions

**Comments to the Author**

1. If the authors have adequately addressed your comments raised in a previous round of review and you feel that this manuscript is now acceptable for publication, you may indicate that here to bypass the “Comments to the Author” section, enter your conflict of interest statement in the “Confidential to Editor” section, and submit your "Accept" recommendation.

Reviewer #1: (No Response)

Reviewer #2: (No Response)

2. Is the manuscript technically sound, and do the data support the conclusions?

Reviewer #1: Yes

Reviewer #2: Partly

3. Has the statistical analysis been performed appropriately and rigorously? 

Reviewer #1: No

Reviewer #2: Yes

4. Have the authors made all data underlying the findings in their manuscript fully available?

Reviewer #1: Yes

Reviewer #2: No

5. Is the manuscript presented in an intelligible fashion and written in standard English?

Reviewer #1: Yes

Reviewer #2: Yes

6. Review Comments to the Author

Reviewer #1: Reviewer 1: AUD subjects were abstinent for a minimum of 4 weeks but an average of 8 years. This wide range of sobriety supports persisting deficits in emotional and memory processing, but also in a heterogenous group, could length of sobriety have contributed to sex differences. Can length of sobriety be used as a co-variate in the analysis?

Authors: We added text to the Limitations portion of the Discussion regarding sobriety with the following: “Regarding the variable of LOS, since there were no values for the NC group, the variable could not be used as a covariate in an analysis of group differences. However, we believe that the alcohol consumption and abstinence characteristics of our AUD cohort are representative of the national population (World Health Organization 2019), thereby improving generalizability of our results.”'

This variable (LOS) be used as a covariate among the AUD group only as a post hoc test to determine if it modulates connectivity.

Reviewer #2: The authors have largely responded to my reviews satisfactorily and I believe this paper makes an interesting contribution to the literature. However, I still have a couple of points of concern regarding the analysis.

i) In their response letter, the authors state that when they chose a fixed arbitrary cluster-size threshold, at least in part because “cluster sizes that have been set by using a cluster-wise p-value may lead to higher underlying false positive rates than those expected when determining cluster size thresholds using Gaussian random field theory (Eklund et al. 2016).”

The authors should consider using mri_glmfit-sim (or an equivalent program) to provide the cluster size threshold at an initial p-value of p=.001 that would be equivalent to a cluster corrected p<.05 or provide the equivalent p-value for a cluster corrected threshold with an initial p-value of p=.001 and a cluster size of 100 mm2. It is my understanding that the faulty assumptions in this procedure leading to the Eklund paper have since been fixed within most neuroimaging software packages (Greve and Fischl, 2017; Cox, Chen, Glen, Reynolds, and Taylor, 2017). Although I am sympathetic to the view that an alpha level of p<.05 is an arbitrary threshold and it should be carefully considered whether or not to actually use an alpha level of p<.05, readers should have some quantitative point of reference for judging the significance of the clusters reported.

ii) With regard to conducting F-tests for cluster-level analyses, the authors state in their response letter that “building an ANOVA model with cluster correction is not as simple as it is for an ROI analyses conducted with traditional statistical software... The cluster extents for the interaction F-tests and the clusters identified for the simple group comparison t-tests would be different, and we know of no neuroimaging software that provides a straightforward means of conducting vertex- and voxel-wise direct group comparisons within an interaction model, with respect to activation clusters.”

I would suggest conducting a whole-brain repeated measures 2 (Group: AUD vs NC) x 2 (Gender: Male vs Female) x 3 (Facial Expression: Positive, Negative, Neutral) ANOVA on the BOLD response data. Then, using each individual significant cluster for each interaction effect (Group-by-Gender, Group-by-Facial Expression, Gender-by-Facial Expression, Group-by-Gender-by-Facial Expression), extract the average parameter estimates for each within-subject condition (in this case, each of the three levels of facial expression) for each subject, and then import these data into SPSS (along with the data for the between-subjects variables) where post-hoc testing for interaction effects within a repeated measures model can be conducted.

I have been able to conduct this procedure successfully in SPM and AFNI in the past; is there not a parallel way to do this in Freesurfer? While this is not a perfect procedure, I believe that it does somewhat control for Type I error since it limits the amount of individual contrasts tested for in the main neuroimaging analysis and it allows for the directionality of the interaction to be probed.

7. PLOS authors have the option to publish the peer review history of their article (what does this mean?). If published, this will include your full peer review and any attached files.

Reviewer #1: No

Reviewer #2: No

---

## [Author Response · Author response to Decision Letter 1]

24 Feb 2021

Please see the attached response to the reviewers.

---

## [Decision Letter · Decision Letter 2]

8 Mar 2021

Brain responsivity to emotional faces differs in men and women with and without a history of alcohol use disorder

PONE-D-20-00671R2

Dear Dr. Sawyer,

We’re pleased to inform you that your manuscript has been judged scientifically suitable for publication and will be formally accepted for publication once it meets all outstanding technical requirements.

Kind regards,

Wi Hoon Jung, PhD

Academic Editor

PLOS ONE

Reviewers' comments:

Reviewer's Responses to Questions

**Comments to the Author**

1. If the authors have adequately addressed your comments raised in a previous round of review and you feel that this manuscript is now acceptable for publication, you may indicate that here to bypass the “Comments to the Author” section, enter your conflict of interest statement in the “Confidential to Editor” section, and submit your "Accept" recommendation.

Reviewer #1: All comments have been addressed

Reviewer #2: All comments have been addressed

2. Is the manuscript technically sound, and do the data support the conclusions?

Reviewer #1: (No Response)

Reviewer #2: Yes

3. Has the statistical analysis been performed appropriately and rigorously? 

Reviewer #1: Yes

Reviewer #2: Yes

4. Have the authors made all data underlying the findings in their manuscript fully available?

Reviewer #1: Yes

Reviewer #2: No

5. Is the manuscript presented in an intelligible fashion and written in standard English?

Reviewer #1: Yes

Reviewer #2: Yes

6. Review Comments to the Author

Reviewer #1: The authors have adequately addressed concerns.

xxxxxxxxxxxxxxxxxxxxxxxxxxxxxxxxxxxxxxxxxxxxxxxxxxx

Reviewer #2: My concerns have been sufficiently addressed and I recommend publication of this manuscript without further revision.

7. PLOS authors have the option to publish the peer review history of their article (what does this mean?). If published, this will include your full peer review and any attached files.

Reviewer #1: No

Reviewer #2: No

---

## [Editor Report · Acceptance letter]

14 May 2021

PONE-D-20-00671R2 

Brain Responsivity to Emotional Faces Differs in Men and Women with and without a History of Alcohol Use Disorder 

Dear Dr. Sawyer:

I'm pleased to inform you that your manuscript has been deemed suitable for publication in PLOS ONE. Congratulations! Your manuscript is now with our production department. 

Kind regards, 

on behalf of

Dr. Wi Hoon Jung 

Academic Editor

PLOS ONE